

# Contextual and individual determinants of oral health-related quality of life among five-year-old children: a multilevel analysis

Monalisa C. Gomes[1,*], Érick T.B. Neves[1], Matheus F. Perazzo[2], Saul M. Paiva[2], Fernanda M. Ferreira[2] and Ana F. Granville-Garcia[1,*]

[1] Departamento de Odontologia, Universidade Estadual da Paraíba, Campina Grande, Paraíba, Brasil
[2] Departamento de Odontopediatria e Ortodontia, Universidade Federal de Minas Gerais, Belo Horizonte, Minas Gerais, Brasil
[*] These authors contributed equally to this work.

## ABSTRACT

**Background**. Contextual factors may influence oral health-related quality of life (OHRQoL) in children. The aim of the present study was to evaluate the influence of individual and contextual determinants of OHRQoL based on the perceptions of children.

**Methods**. A cross-sectional study was conducted with a representative sample of 769 five-year-old children from public and private preschools in a city in the countryside of northeast Brazil. Parents/caregivers answered questionnaires addressing psychological aspects, sociodemographic data and aspects of the child's oral health. The children answered the child version of the Scale of Oral Health Outcomes for five-year-old children and were submitted to oral examinations. Variables related to the context were obtained from the schools and official municipal publications. Unadjusted and adjusted multilevel Poisson regression models were used to investigate associations between variables.

**Results**. In the adjusted analysis, parent's/caregiver's schooling, household income, parent's/caregiver's age, a history of dental pain, dental caries and its consequences and traumatic dental injury were considered individual determinants of OHRQoL according to the children's self-reports. After the incorporation of the contextual determinants, the association between parent's/caregiver's schooling and OHRQoL lost its significance. Type of school was the context variable that remained associated with OHRQoL.

**Discussion**. Besides the clinical and sociodemographic individual characteristics, characteristics of the school environment in which the child studies are associated with self-reported impacts on OHRQoL.

Corresponding author
Ana F. Granville-Garcia,
anaflaviagg@ccbs.uepb.edu.br,
anaflaviagg@hotmail.com

## INTRODUCTION

The oral health status of preschool children has been the object of diverse studies due to the high prevalence of oral problems in this age group, such as dental caries, traumatic dental injury (TDI) and malocclusion (*Kramer et al., 2013*; *Gomes et al., 2014*). Moreover, these conditions can cause impacts on oral health-related quality of life (OHRQoL) among children and their families (*Kramer et al., 2013*; *Gomes et al., 2014*; *Abanto et al., 2014*). Psychological aspects of parents/caregivers, such as sense of coherence (SOC) and locus of control (LOC), may also be related to oral health problems and OHRQoL (*Bonanato et al., 2009a*; *Khatri, Acharya & Srinivasan, 2014*; *Gururatana, Baker & Robinson, 2014*; *Fernandes et al., 2017*; *Nunes & Perosa, 2017*).

OHRQoL is a multidimensional concept that reflects functional, psychological and social aspects (*Tsakos et al., 2012*). Most studies involving preschool children have evaluated OHRQoL using two previously validated questionnaires. The Early Childhood Oral Health Impact Scale (ECOHIS) was the first questionnaire to emerge for the evaluation of OHRQoL among children aged two to five years and their families based on the reports of parents/caregivers (*Pahel, Rozier & Slade, 2007*). More recently, the Scale of Oral Health Outcomes for 5-year-old children (SOHO-5) was developed for the evaluation of the OHRQoL of five-year-olds (*Tsakos et al., 2012*; *Abanto et al., 2013a*), which considers the perceptions of parents/caregivers as well as the perceptions of children (self-report). A previous study reports that this questionnaire has good psychometric properties and a good correlation is found between the two versions (parental and child) (*Abanto et al., 2013b*). Thus, this scale enables a better evaluation of the impact of oral health conditions on the OHRQoL of five-year-old children.

Previous investigations conducted with preschool children have demonstrated the impact of dental caries, TDI and malocclusion on the OHRQoL of children (*Wong et al., 2011*; *Kramer et al., 2013*; *Gomes et al., 2014*; *Abanto et al., 2014*; *Onoriobe et al., 2014*; *Arrow & Klobas, 2015*; *Guedes et al., 2014*; *Perazzo et al., 2017b*; *Abanto et al., 2018*). Most studies focused on the evaluation of the characteristics of the children and their families for the determination of OHRQoL (*Wong et al., 2011*; *Kramer et al., 2013*; *Gomes et al., 2014*; *Abanto et al., 2014*; *Onoriobe et al., 2014*; *Arrow & Klobas, 2015*; *Perazzo et al., 2017b*; *Abanto et al., 2018*). However, it is possible to find explanations for the impact on OHRQoL not only in individual characteristics, but also in contextual characteristics. Only one study that evaluated the impact of both individual and contextual factors on the OHRQoL of preschool children was found (*Guedes et al., 2014*). The study investigated the perceptions of parents using the ECOHIS and found that unfavorable social conditions have a negative impact on the reports of parents/caregivers regarding the OHRQoL of children, but no investigation was performed on the impact of the school environment, which exerts considerable influence in this phase of one's life as well as on the OHRQoL of children (*Guedes et al., 2014*). While studies evaluating contextual aspects of OHRQoL have involved different age groups (*Turrell et al., 2007*; *Alwadi & Vettore, 2017*; *Bomfim, Herrera & De-Carli, 2017*), no previous studies have evaluated the influence of individual and contextual factors on the self-reports of five-year-old children regarding OHRQoL.

Thus, there is a need for a clearer understanding of the impact of oral conditions based on the self-reports of children.

Contextual factors seem to be strongly associated with different oral health outcomes (*Fernández et al., 2015*; *Piovesan et al., 2017*) and the study of these factors is undoubtedly important to the planning of services as well as the investigation of health inequalities (*Petersen & Kwan, 2011*). The school setting is important to the intellectual development of children and also exerts an influence on health behaviors. Moreover, there is a relationship between economic status and the type of preschool a child attends (*Piovesan et al., 2017*). Thus, understanding these contextual disparities could be the basis for directed interventions and health policies. The use of statistical methods for a multilevel analysis assists in better data treatment because the findings begin to demonstrate a hierarchical structure (*Diez-Roux, 2000*).

Thus, the aim of the present study was to evaluate the influence of individual and contextual determinants on OHRQoL according to the perceptions of five-year-old children.

## MATERIALS AND METHODS

### Sample and study design

The paper is based on a previously published doctoral thesis (*Gomes, 2017*). Moreover, this study followed the same methodology used in previous studies (*Perazzo et al., 2017a*; *Perazzo et al., 2017b*; *Gomes et al., 2018*). The present study was conducted at public and private preschools between August and December 2015 in the city of Campina Grande, which is located in northeast Brazil, with approximately 400,000 inhabitants.

The sample was selected using a complex two-stage (preschools and children) probabilistic sampling method. Two hundred sixty-three preschools (129 public and 134 private) were registered with the Ministry of Education. The city is divided into six administrative districts and preschools were randomly selected proportional to the total number in each district. Twenty-eight private and 20 public preschools were selected. In the second stage, five-year-old children enrolled in these preschools were randomly selected using a simple lottery procedure.

The sample size was calculated considering a 5% margin of error, 95% confidence interval and a 1.6 design effect to account for the change in the precision of the estimates due to the two-stage sampling process. Moreover, a 50% prevalence of the negative impact on OHRQoL was used to maximize the sample size and enhance statistical power of the results. The minimum sample was determined to be 615 children. This value was increased to compensate for possible dropouts estimated at 20% resulting in a sample of 769 five-year-old children.

### Eligibility criteria

Children aged five years attending public and private preschools were included in the study. Those with a systemic adverse health conditions according to the reports of parents/caregivers, those with permanent teeth and those having been submitted to orthodontic treatment were excluded from the study.

## Calibration exercises

Two researchers who performed the data collection and an experienced specialist in the field participated in this phase. The researchers first evaluated photographs of oral conditions and group discussions were held. In the clinical phase, 40 children were randomly selected from a preschool that did not participate in the main study. These children were examined twice. The first examination was used for the calculation of inter-examiner agreement (Kappa statistic) between the researchers and the experienced specialist ($K = 0.89 - 0.90$ for dental caries; $K = 0.90 - 1.00$ for the pufa index (consequences of untreated dental caries); $K = 0.88 - 0.90$ for TDI; $K = 0.86 - 0.91$ for malocclusion and $K = 0.68 - 0.73$ for tooth wear). After a seven-day interval, the same children were examined a second time for the calculation of intra-examiner agreement ($K = 0.87 - 1.00$ for dental caries; $K = 1.00$ for the pufa index; $K = 0.82 - 0.87$ for TDI; $K = 0.94 - 1.00$ for malocclusion and $K = 0.81 - 1.00$ for tooth wear). The Kappa coefficients demonstrated good reliability for the clinical examinations, as coefficients between 0.61 and 0.80 are considered good and those between 0.81 and 1.00 are considered very good (*Altman, 2006*).

## Data collection

Data collection was performed at the previously selected preschools following contact with the principals of each preschool to explain the study and dynamics of the data collection process. Parents/caregivers were then asked to participate in a meeting at their child's preschool for clarifications regarding the objectives of the study and obtain written consent for the examination of the children. At the same meeting, the parents/caregivers were asked to fill out the questionnaires. The children answered the SOHO-5 questionnaire prior to the clinical exams. After the questionnaires were collected, the children were examined for the assessment of the oral conditions.

### *Individual determinants*

*Individual sociodemographic variables.* To obtain an individual profile of each child/family, the following sociodemographic data were collected: child's sex, parent's/caregiver's schooling, parent's/caregiver's age (in years), household income (analyzed based on the Brazilian monthly minimum wage, which was equivalent to U$280 at the time of the data collection) and whether the child had siblings.

*Oral health-related variables.* Some aspects of the children's oral health were collected from the parents/caregivers. A history of dental pain was recorded if this symptom was reported/observed at least once in the child's life. Visit to the dentist was recorded if this occurred sometime in the child's life, independently of the reason. Tooth brushing frequency was investigated and dichotomized as <2 times a day or ≥ 2 times a day.

*Sense of coherence.* The SOC of the parents/caregivers was measured using the Sense of Coherence Scale (SOC-13), employing the version validated in Brazil for use on mothers of preschool children (*Bonanato et al., 2009b*). This questionnaire has 13 items, each with five response options that assist in evaluating the components that compose SOC: comprehensibility, manageability and meaningfulness. The total ranges from 13 to 69

points, with higher scores indicative of a stronger SOC and greater capacity to cope with stress. For the purposes of statistical analysis, the score was dichotomized by the median, as performed in a previous study (*Bonanato et al., 2009a*). Scores below the median were considered indicative of a weak SOC and scores above the median were indicative of a strong SOC.

*Locus of control.* The LOC of the parents/caregivers was evaluated using the Multidimensional Health Locus of Control (version validated in Brazil) (*Nunes & Perosa, 2017*), which has 18 items distributed among three subdivisions (internal/external/chance) for the evaluation of the respondent's perception of who or what determines health/illness events: the individual himself/herself (internal) or other forces (external/chance). Each item has five response options (1 = fully agree; 2 = agree in part; 3 = neither agree nor disagree; 4 = disagree in part; 5 = fully disagree). The scores of the items on each subscale are totaled and can range from 6 to 30 points, with higher scores on the subscale indicating a lower degree of each factor (internal and external/chance). An internal LOC is considered when the lowest score is on the subscale of internal factors and an external LOC is considered when the lowest score is on the subscale of external or chance factors.

*Clinical examination.* The clinical examinations were performed at the preschools in the knee-to-knee position. The children were first given a kit with a toothbrush, toothpaste and dental floss and then performed oral hygiene under the supervision of the researchers. The examiners used individual protective equipment and a head lamp (Petzl Zoom head lamp; Petzl America, Clearfield, UT, USA). A sterilized mouth mirror (PRISMA, São Paulo, SP, Brazil), sterilized Williams probe (WHO-621; Trinity, Campo Mourão, PR, Brazil) and gauze were used. The clinical examinations were performed using criteria established in the literature. After the examination, a fluoride varnish (Duraphat®—5% NaF) was applied to the teeth and the researchers sent a letter to the parents/caregivers informing them of their child's oral health status.

Dental caries was evaluated using the International Caries Detection and Assessment System (ICDAS-II) (*Ismail et al., 2007*). The score ranges from 0 to 6. Code 0 refers to a sound tooth. Code 1 refers to a white spot detected after drying the teeth with compressed air. Code 2 refers to a white spot diagnosed following drying of the teeth with gauze. Codes 3 to 6 are used for increasing degrees of cavitated lesions. In the present study, the children were classified in three dental caries categories coded as follows: absent (ICDAS code 0); white spot (children with caries only in the initial stage [ICDAS code 2]); and cavitated lesion (children with at least one cavitated tooth [ICDAS code 3 to 6]). Code 1 is not used in epidemiological studies since the teeth are dried with gauze in such studies rather than compressed air.

Caries activity was also evaluated. Enamel lesions were recorded as active using the following criteria: lesion is whitish/yellowish; lesion is chalky (lack of luster); lesion may or may not be cavitated; lesion feels rough upon probing; probing may or may not encounter cavity. Dentin lesions were recorded as active using the following criteria: lesion may appear

as shadow below intact, but de-mineralized enamel; if cavity extends into dentin, dentin appears yellowish/brownish; dentin soft upon probing (*Pitts, 2009*).

The pufa index was used to evaluate the consequences of untreated dental caries in the children (*Monse et al., 2010*): visible pulpal involvement (p), ulceration (u) caused by dislocated tooth fragments, fistula (f) and abscess (a). In the present study, this variable was dichotomized as absent (no consequences of untreated caries) or present (one or more teeth with some consequence of untreated caries).

The determination of TDI was based on the criteria established by the literature in enamel fracture, enamel + dentin fracture, complicated crown fracture, luxation (lateral, intrusive and extrusive) and avulsion (*Andreasen, Andreasen & Andersson, 2007*). Discoloration stemming from trauma was also investigated. This variable was dichotomized as absent or present (one or more teeth diagnosed with some type of TDI or discoloration stemming from a trauma).

For the evaluation of malocclusion, the following types were investigated: increased overbite (>2 mm), increased overjet (>2 mm), anterior open bite, anterior crossbite and posterior crossbite (*Foster & Hamilton, 1969*). Malocclusion was recorded as present when a child exhibited at least one of these conditions.

The children were also submitted to a clinical examination for the determination of tooth wear due to attrition. This type of tooth wear is associated with functional and parafunctional habits, such as chewing and bruxism (clenching and/or grinding the teeth). Attrition generally occurs on occlusal, incisal or palatal surfaces on maxillary teeth or vestibular surfaces on mandibular teeth, presenting as a small polished area on the tip of a cusp, in the region around the cusp or on the incisal angles. This oral condition was diagnosed in the presence of wear on the incisal surfaces of the anterior teeth and/or occlusal surfaces of the posterior teeth.

### Contextual determinants

*Contextual variables.* Four variables were investigated for the evaluation of the contextual aspects of OHRQoL: type of preschool (public or private), number of children in the preschool (school size), mean monthly income of the neighborhood in which the preschool was located and number of oral health teams in the administrative district in which the school was located. Information on the mean income of the neighborhood was obtained from the Brazilian Institute of Geography and Statistics in the city and the number of oral health teams in the administrative districts was obtained from the Ministry of Health in the city. Data on the preschools were recorded during the first visit to each preschool.

### Outcome

*Oral health-related quality of life.* Quality of life was evaluated using the Scale of Oral Health Outcomes for 5-year-old children (SOHO-5), which is a validated questionnaire in Brazil for the evaluation of the impact of oral problems on the OHRQoL of children aged five years (*Tsakos et al., 2012*; *Abanto et al., 2013a*) and is divided into two versions: child version and parental version. As the aim of the present study was to evaluate self-reports of children with regard to OHRQoL, only the child version of the SOHO-5 was considered. The questionnaire addresses difficulty eating, difficulty speaking, difficulty playing, difficulty

sleeping, the avoidance of smiling due to pain, the avoidance of smiling due to appearance and difficulty drinking. The answers are scored on a three-point scale (no = 0, a little = 1 and a lot = 2). To facilitate the child's responses, a self-explanatory drawing for each type of answer was used. The sum of all answers is used for the final score, which ranges from 0 to 14 points on the child version of the questionnaire.

## Statistical analysis

Descriptive statistics were used for the characterization of the sample. Unadjusted and adjusted multilevel Poisson regression models were created to describe associations between the outcomes and predictors. The sum of the scores on the SOHO-5 was considered for the evaluation of OHRQoL. Multilevel Poisson regression analysis involved a fixed effects model with random intercepts to evaluate associations between mean total SOHO-5 (dependent variable) and both individual and contextual covariates (independents variables). The sum of all answers ranges from 0 to 14 points on the child version of the questionnaire. This strategy enabled the estimation of rate ratios (RR) between comparison groups and respective 95% confidence intervals (CI).

In the first step, an unconditional (null) model was used to estimate the variability in the data before the individual and contextual characteristics were taken into account (*Diez-Roux, 2000*). Individual and contextual variables that achieved a *p*-value <0.20 in the univariate multilevel Poisson regression analysis were incorporated into the multiple model and those with a *p*-value <0.05 in the adjusted analysis remained in the model. Interactions between individual covariates and contextual variables in the model were tested and those with statistical significance ($p < 0.05$) were incorporated into the final model. The models were also tested for multicollinearity and no collinearity was found between the individual and contextual factors (variance inflation factor <2, tolerance statistic >0.6 and correlation coefficients <0.5 between all possible combinations of variables). The goodness-of-fit of the models was calculated based on deviance values (−2 log likelihood).

Statistical analyses were performed using the Statistical Package for the Social Sciences (SPSS for Windows, version 19.0, SPSS Inc., Chicago, IL, USA) and Hierarchical Linear and Nonlinear Modeling (HLM 6.06 statistical package). SPSS was used for the descriptive analyses and to create two databases using individual and contextual variables. These data bases were then used in the HLM 6.06 statistical package to perform multilevel analyses.

## Ethical aspects

This study received approval from the Human Research Ethics Committee of the State University of Paraíba (38937714.0.0000.5187) and was conducted in compliance with the guidelines stipulated in the Declaration of Helsinki. All legal guardians signed a statement of informed consent prior to the data collection process.

## RESULTS

A total of 769 pairs of children and parents/caregivers participated in the study. The male sex accounted for 52.4% of the sample, 30.0% of the parents/caregivers had eight years of schooling or less and the majority of the children (65.7%) had siblings. Regarding
**Table 1** Individual and contextual characteristics of sample.

| Variable | Type of preschool | |
|---|---|---|
| | Public | Private |
| | *n*(%)/mean (SD) | |
| **Individual Level** | | |
| Sex | | |
|    Female | 136(45.6) | 230(48.8) |
|    Male | 162(54.4) | 241(51.2) |
| Parent's/caregiver's schooling[a] | | |
|    ≤8 years of study | 184(62.2) | 46(9.8) |
|    >8 years of study | 112(37.8) | 424(90.2) |
| Monthly household income[a] | 774.71(456.39) | 2,587.29(3,108.16) |
| Parent's/caregiver's age[a] | 32.63(8.98) | 32.69(7.00) |
| Only child[a] | | |
|    No | 228(77.6) | 273(58.2) |
|    Yes | 66(22.4) | 196(41.8) |
| History of dental pain[a] | | |
|    No | 184(63.0) | 392(84.5) |
|    Yes | 108(37.0) | 72(15.5) |
| Dental caries | | |
|    Absent | 10(3.4) | 57(12.1) |
|    White spot | 55(18.5) | 195(41.4) |
|    Cavitated lesion | 233(78.2) | 219(46.5) |
| Consequence of untreated dental caries (visible pulpal involvement, ulceration, fistula and/or abscess) | | |
|    Absent | 228(76.5) | 435(92.4) |
|    Present | 70(23.5) | 36(7.6) |
| TDI | | |
|    Absent | 141(47.3) | 222(47.1) |
|    Present | 157(52.7) | 249(52.9) |
| Malocclusion[a] | | |
|    Absent | 126(42.4) | 199(42.3) |
|    Present | 171(57.6) | 272(57.7) |
| Tooth wear | | |
|    Absent | 68(22.8) | 104(22.1) |
|    Present | 230(77.2) | 367(77.9) |
| Sense of coherence | | |
|    Weak | 168(56.4) | 153(32.5) |
|    Strong | 130(43.6) | 318(67.5) |
| Locus of control[a] | | |
|    Internal | 171(58.2) | 349(74.3) |
|    External | 123(41.8) | 121(25.7) |

| Variable | Type of preschool | |
|---|---|---|
| | **Public** | **Private** |
| | *n*(%)/mean (SD) | |
| **Contextual level** | | |
| Mean monthly income of neighborhood | 1,074.77(500.97) | 996.19(454.38) |
| Number of oral health teams | 7.97(1.03) | 9.55(2.40) |
| Number of children in preschool | 80.86(53.74) | 124.63(116.18) |

**Notes.**

[a] Sample less than 769 due to the failure of some interviewees to provide this information or impossibility of performing the clinical examination.

characteristics related to oral health, the majority of children who attended public preschools had cavitated lesions (78.2%), consequences of untreated dental caries (23.5%) and a history of dental pain (37.0%). In contrast, TDI (52.9%), malocclusion (57.7%) and tooth wear (77.9%) were more prevalent among the children attending private preschools (Table 1).

Table 2 displays the results of the multilevel Poisson regression analysis. Significant associations were found in the univariate analysis for the following variables: parent's/caregiver's schooling (RR = 1.31; 95% CI [1.15–1.49]), household income (RR = 0.93; 95% CI [0.91–0.96]), parent's/caregiver's age (RR = 1.01; 95% CI [1.01–1.02]), being an only child (RR = 0.87; 95% CI [0.78–0.97]), history of dental pain (RR = 1.69; 95% CI [1.52–1.88]), white spot (RR =1.39; 95% CI [1.12–1.72]), cavitated lesion (RR =1.69; 95% CI [1.37–2.08]), caries activity (RR = 1.98; 95% CI [1.62–2.42]), consequence of untreated dental caries (RR = 1.51; 95% CI [1.33–1.71]), TDI (RR = 1.19; 95% CI [1.08–1.31]), tooth wear (RR = 1.13; 95% CI [1.01–1.27]) and type of preschool (RR = 2.10; 95% CI [1.64–2.70]).

After adjusting for the individual and contextual variables, household income (RR =0.86; 95% CI [0.82–0.91]), parent's/caregiver's age (RR =1.01; 95% CI [1.01–1.02]), history of dental pain (RR =1.55; 95% CI [1.37–1.76]), white spot (RR =1.45; 95% CI [1.14–1.85]), cavitated lesion (RR =1.43; 95% CI [1.13–1.82]), consequence of untreated dental caries (RR =1.22; 95% CI [1.06–1.40]) and TDI (RR =1.20; 95% CI [1.08–1.33]) were identified as individual determinants of a negative impact on the OHRQoL of the children based on self-reports. After the incorporation of the contextual variables (type of preschool), parent's/caregiver's schooling lost its statistical significance. The influence of the type of preschool is demonstrated on the contextual level, as children who attended public preschools reported a greater impact on OHRQoL (RR =1.95; 95% CI [1.51–2.54]). There was no collinearity between individual and contextual factors (variance inflation factor <2, tolerance statistic >0.6 and correlation coefficients <0.5 between all possible combinations of variables). On the other hand, type of preschool demonstrated significant interactions with parent's/caregiver's schooling [RR 1.40 (1.03–1.89)], household income [RR 0.79 (0.69–0.90)] and parent's/caregiver's age [RR 1.02 (1.01–1.03)]. Therefore, these interaction terms were retained in the final model for adjustment.

**Table 2** Unadjusted and adjusted assessment of the association between overall SOHO-5 scores and both individual and contextual variables.

| Variable | SOHO-5 | | |
| --- | --- | --- | --- |
| | Scores mean (SD) | Crude[a] RR (95% CI) | Adjusted[b] RR (95% CI) |
| **Individual Level** | | | |
| Sex | | | |
|     Female | 2.20(3.22) | 0.92(0.84–1.01) | – |
|     Male | 2.53(3.45) | 1.00 | – |
| Parent's/caregiver's schooling | | | |
|     ≤8 years of study | 3.53(4.03) | **1.31(1.15–1.49)** | 0.95(0.81–1.12) |
|     >8 years of study | 1.88(2.88) | **1.00** | 1.00 |
| Household income (MMW) | – | **0.93(0.91–0.96)** | **0.86(0.82–0.91)** |
| Parent's/caregiver's age | – | **1.01(1.01–1.02)** | **1.01(1.01–1.02)** |
| Only child | | | |
|     Yes | 1.87(2.97) | **0.87(0.78–0.97)** | – |
|     No | 2.61(3.47) | **1.00** | – |
| Use of dental services | | | |
|     Yes | 2.25(3.22) | 1.02(0.93–1.13) | – |
|     No | 2.47(3.45) | 1.00 | – |
| Tooth brushing frequency | | | |
|     <Twice a day | 2.90(3.47) | 1.09(0.94–1.26) | – |
|     ≥Twice a day | 2.32(3.34) | 1.00 | – |
| History of dental pain | | | |
|     Yes | 3.77(3.88) | **1.69(1.52–1.88)** | **1.55(1.37–1.76)** |
|     No | 1.92(3.01) | **1.00** | 1.00 |
| Dental caries | | | |
|     Absent | 1.22(2.04) | **1.00** | 1.00 |
|     White spot | 1.84(2.87) | **1.39(1.12–1.72)** | **1.45(1.14–1.85)** |
|     Cavitated lesion | 2.84(3.66) | **1.69(1.37–2.08)** | **1.43(1.13–1.82)** |
| Caries activity | | | |
|     Active | 2.68(3.53) | **1.98(1.62–2.42)** | – |
|     Inactive | 1.18(2.31) | **1.00** | – |
| Consequence of untreated caries (visible pulpal involvement, ulceration, fistula and/or abscess) | | | |
|     Absent | 2.15(3.22) | **1.00** | 1.00 |
|     Present | 3.79(3.79) | **1.51(1.33–1.71)** | **1.22(1.06–1.40)** |
| TDI | | | |
|     Absent | 2.05(3.25) | **1.00** | 1.00 |
|     Present | 2.67(3.41) | **1.19(1.08–1.31)** | **1.20(1.09–1.33)** |
| Malocclusion | | | |
|     Absent | 2.31(3.29) | 1.00 | – |
|     Present | 2.42(3.40) | 1.02(0.92–1.12) | – |

**Table 2** (*continued*)

| Variable | SOHO-5 | | |
|---|---|---|---|
| | Scores mean (SD) | Crude[a] RR (95% CI) | Adjusted[b] RR (95% CI) |
| Tooth wear | | | |
|     Absent | 2.22(3.42) | **1.00** | – |
|     Present | 2.42(3.33) | **1.13(1.01–1.27)** | – |
| Sense of coherence | | | |
|     Weak | 2.76(3.56) | 1.00 | – |
|     Strong | 2.10(3.16) | 0.94(0.85–1.04) | – |
| Locus of control | | | |
|     Internal | 2.21(3.19) | 0.94(0.85–1.04) | – |
|     External | 2.70(3.63) | 1.00 | – |
| **Contextual Level** | | | |
| Type of school | | | |
|     Public | 3.54(4.05) | **2.10(1.64–2.70)** | **1.95(1.51–2.54)** |
|     Private | 1.63(2.56) | **1.00** | **1.00** |
| Mean monthly income of neighborhood | – | 0.99(0.99–1.00) | – |
| Number of oral health teams | – | 0.96(0.88–1.04) | – |
| Number of children in preschool | – | 0.99(0.99–1.00) | – |

**Notes.**
[a] Univariate multilevel Poisson regression models.
[b] Multiple multilevel Poisson regression model adjusted by individual and contextual variables presented in table as well as following interactions: "parent's/caregiver's schooling and type of preschool" [RR 1.40 (1.03–1.89)], "household income and type of preschool" [RR 0.79 (0.69–0.90)] and "parent's/caregiver's age and type of preschool" [RR 1.02 (1.01–1.03)].
MMW, Brazilian monthly minimum wage (equivalent to U$280 at time of data collection).

# DISCUSSION

This study was conducted to evaluate the influence of individual and contextual determinants on OHRQoL based on self-reports by children. To the best of our knowledge, this is the first study with this objective. The main findings demonstrate that individual socioeconomic factors and clinical conditions exert an influence on this perception. In the final model, OHRQoL was associated with household income, parent's/caregiver's age, a history of dental pain, dental caries, consequences of untreated dental caries and TDI. However, the results provide evidence that the social context also exerts an influence on OHRQoL, such as the social environment of the school at which children study. Children who attended public schools demonstrated greater impact on OHRQoL than those who attended private schools.

Clinical conditions, such as dental caries and its consequences (pulpal involvement, ulceration, fistula and abscess) and TDI were associated with OHRQoL according to self-reports by the children, even after adjusting for contextual variables. Previous studies have also demonstrated such associations, reporting that these conditions may be related to impairments with regard to functional, esthetic and social aspects (*Kramer et al., 2013*; *Gomes et al., 2014*; *Abanto et al., 2014*; *Guedes et al., 2014*; *Perazzo et al., 2017b*; *Abanto et al., 2018*). It is also possible that these relationships were due to the associated symptoms. A history of dental pain was associated with OHRQoL, which is in agreement

with data from previous studies involving this age group (*Clementino et al., 2015*). However, white spots were also associated with OHRQoL. It is possible that children with white spots on their anterior teeth may perceive a negative esthetic effect. This finding is also described in a previous study, although no such association was found in the perception of parents/caregivers (*Perazzo et al., 2017b*). This may demonstrate that parents/caregivers perceive oral problems in small children only in the presence of pain, which underscores the importance of studies that also evaluate children's perceptions to gain a better understanding of OHRQoL in young children. Moreover, malocclusion was not associated with the children's perceptions regarding OHRQoL, which may reflect the absence of symptoms with this type of oral problem as well as the greater prevalence of mild malocclusions, which would not result in a negative perception on the part of the children regarding oral esthetics.

Parent's/caregiver's schooling, household income and parent's/caregiver's age were individual determinants of OHRQoL according to self-reports by the children. However, after adjusting for contextual variables, only household income and parent's/caregiver's age remained in the final model. This finding is in agreement with data reported in a previous study, which found that a lower income and younger age of parents/caregivers exerted a greater impact on OHRQoL (*Guedes et al., 2014*). Individuals with a lower income have less access to health services and information and younger parents/caregivers generally have less experience with regard to health care (*Martins-Júnior et al., 2013*). However, the influence of these individual factors should be considered with caution, as such factors had a significant interaction with the contextual variable in the model, demonstrating that the effect on OHRQoL was different for children who attended public and private schools. Moreover, type of preschool seems to have a greater influence on the oral health of children than parent's/caregiver's schooling.

Type of preschool was the contextual variable that remained in the final model. Children attending public preschools reported worse OHRQoL, which may demonstrate a lower socioeconomic status of the family. In Brazil, most children and adolescents who study at public schools are from underprivileged families that live in areas of social deprivation and do not have the financial resources to afford a private school (*Piovesan et al., 2011*). Besides this individual socioeconomic issue, public and private preschools in Brazil have very distinct characteristics, which may explain the results of the present study. A previous study involving preschool children demonstrated that those who find themselves in an unfavorable social context also have poorer OHRQoL according to the reports of parents/caregivers (*Guedes et al., 2014*). It is possible that the other contextual variables analyzed were not associated because they demonstrated a division with a geographical element (such as mean income of the neighborhood), since individuals with different socioeconomic statuses often reside in the same area. Thus, the influence of characteristics related to type of school on the OHRQoL of children should be explored further in future studies.

Schools are considered important settings for the promotion of health in children (*Piovesan et al., 2011*), since children spend a large part of their time in such environments. Thus, schools could be used for measures designed to improve health, self-esteem

and healthy behaviors in children (*Fernández et al., 2015*). Indeed, the present study demonstrates that the school setting could be a good option for the planning of preventive strategies directed at oral problems and, consequently, reduce the negative impact on OHRQoL. It is possible that some schools, besides having children with a better economic status, also develop activities that stress the importance of health care. In a study involving children aged one to five years, children whose mothers had a greater participation in their children's schools were more likely to make use of oral health services (*Piovesan et al., 2017*). Thus, health programs for children should consider the school environment. Oral health programs can assist in the implementation of preventive measures, such as healthy behaviors.

With regard to psychological aspects of the parents/caregivers, neither SOC nor LOC were associated with OHRQoL. SOC is the capacity to adapt to stress, which may be reflected in the oral health of individuals and their children (*Bonanato et al., 2009b*). Mothers with a low SOC (little capacity to adapt to stressful situations) tend to have children with more caries (*Bonanato et al., 2009a*) and take their children to the dentist less often (*Perazzo et al., 2017a*). LOC is the perception one has regarding who or what controls events in life (the individual himself/herself or others) (*Nunes & Perosa, 2017*). Individuals who believe that they have a greater influence over their own lives (internal locus) tend to have more positive attitudes regarding health. These findings may be due to the fact that OHRQoL was evaluated based on the children's perceptions. Previous studies involving preschool children report an association between a strong SOC on the part of parents and a better OHRQoL according to the perceptions of parents/caregivers (*Khatri, Acharya & Srinivasan, 2014*; *Fernandes et al., 2017*). Another study reports a relationship between the LOC and OHRQoL, but involved students aged 11 to 14 years (*Gururatana, Baker & Robinson, 2014*). Thus, these aspects need to be explored further. The present results demonstrate that socioeconomic and clinical factors are more important to the evaluation of OHRQoL based on the perceptions of children.

The cross-sectional design can be considered a limitation of the present study due to the inability to evaluate causality. However, studies with this type of design are important for estimating the prevalence of events in a representative sample of the population. Moreover, cross-sectional studies provide useful data for the planning of public health policies. As a representative sample and validated questionnaires were employed, the results can be extrapolated to the population of Brazilian five-year-old preschoolers. Further studies are needed to evaluate the longitudinal aspects of the associations found herein.

## CONCLUSION

Type of preschool was identified as a contextual determinant in this study. Moreover, children with a history of dental pain, caries and its consequences and TDI who belonged to families with a lower income and whose parents/caregivers were younger reported poorer OHRQoL.

### Funding

The authors received no funding for this work.

### Competing Interests

The authors declare there are no competing interests.

### Author Contributions

- Monalisa C. Gomes conceived and designed the experiments, performed the experiments, prepared figures and/or tables, authored or reviewed drafts of the paper, approved the final draft.
- Érick T.B. Neves and Matheus F. Perazzo conceived and designed the experiments, performed the experiments, authored or reviewed drafts of the paper, approved the final draft.
- Saul M. Paiva and Ana F. Granville-Garcia conceived and designed the experiments, performed the experiments, analyzed the data, prepared figures and/or tables, authored or reviewed drafts of the paper, approved the final draft.
- Fernanda M. Ferreira analyzed the data, prepared figures and/or tables, authored or reviewed drafts of the paper, approved the final draft.

### Human Ethics

The following information was supplied relating to ethical approvals (i.e., approving body and any reference numbers):

This study received approval from the Human Research Ethics Committee of the State University of Paraíba (38937714.0.0000.5187) and was conducted in compliance with the guidelines stipulated in the Declaration of Helsinki. All legal guardians signed a statement of informed consent prior to the data collection process.

### Data Availability

The raw data are provided in Data S1.

### Supplemental Information

Supplemental information for this article can be found online at http://dx.doi.org/10.7717/peerj.5451#supplemental-information.

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
