# Peer review of "Contextual and individual determinants of oral health-related quality of life among five-year-old children: a multilevel analysis"

_PeerJ, doi:10.7717/peerj.5451_

## Round 0.1 · original submission · Major Revisions

Both reviewers pointed out that this is a interesting studies. Yet there is room for improvement. I hope the reviewers' comments will help you revising your manuscript. I want to enlighten a few points.

Reviewer 1 did important suggestions that could improve your manuscript. To me the Introduction is not too long, but it is a major concern, as pointed out by the referee, that nearly all references refer to Brazilian studies. Please provide a broader view on the topic.

The setting (private and public schools) and country are not clear from the abstract, please add.

The pilot study has no relevance for this paper so you could take this part out.

Finally, when I checked your manuscript for plagiarism I found a similarity Index of 33%. The reason seems to be that your study is part of a Master thesis that is in the repository of your university. You should have mentioned that in your submission letter and it must be stated in the manuscript that the article is based on a published Master thesis.

Reviewer 1 ·

Basic reporting

Except for minor points the english language used is clear and unambiguos.
the article structure is fine. However,I have some suggestions for the improvement of the article:
Introduction:
1. The introduction is too long. An introduction should not exceed one page with 1.5 spacing.
Line 57: clinical conditions is very general,... do you mean oral diseases?
Line 59: The sentence about SOC is completely unclear, unspecific. Why and how is stress reflected in the oral health of an individual?
Line 61: which control of life events is related to oral health? to vague formulated.
Method
The part on Data collection should be sub-divided into individual determinants, contextual determinants and outcome.
Line 135: Why is the pilot study relevant for this research(-question)? Did you use for the calibration of researchers? No? why do you mention it, it does not add anything to this study...
Line 156: do you mean 'anytime'? or at least once? Unclear!
Line 204: code 1 is not used in epidemiological studies. This is a very common sentence... What did you do in your study?

Results:
The prevalences of oral health conditions (except for malocclusion) appear very high for 5-year old children. Table 1 could be presented stratified by private/public schools....This would already point towards the main result of the study and would make the data more transparant for the reader.

The ES for houshold income appears odd. How did you measure houshold income? Did you use it continously or categorical?You don't describe it at any point in your study, although you mention it as one of the most important determinants. You also describe it differently in model 1 and model 2 (monthly houshold income vs lower household income). Please stay consistent.

Tables:
The legends of the tables need to be more specific about which covariates are used in the model

I suppose you mean CI instead of IC?

Dont present intercept, this is not relevant information for the research question

Dont present Deviance in the table, this is not relevant information for the research question

What is the difference between model 1 and the RR given in table 2? Table 2 and 3 should be combined into one table

References

More diverse articles on the topic should be included, for example also from Europe. There are defenitely articles on the association of individual/environmental factors related to OHRQOL in children. Now only articles from Brazil are included... many based on the same data/from the same research groups....

Experimental design

Generally well done. However some suggestions/comments/questions:

Sampling: There are more private than public school registered with the ministry of education. Why in the end do you include more public than private schools in your study? Do you put your study at risk for selection bias in this way? Make this clear in table 1 and eventually discuss this in the limitations of your study.

Sample size: How did you derive at a 1.6 design effect? And what do you mean with a 50% prevalence of the impact on OHRQOL? Any impact? How many points under maximum? Unclear!

Statistical analysis: What about the correlations between individual and contextual factors? Have you test it? How did you deal with it?
have you tested for interaction between all covariates/ determinants and type of school? What were the results?

Validity of the findings

Generally good. However based on the comments above and some extra analysis they should (eventually) be adjusted. For now the validity of the findings are not yet sufficiently supported.

Additional comments

Nice and interesting study!
general comments:
1. In a sentence always first mention determinant and then outcome. In this way it becomes much easier to follow for the reader.
2. Why do you spent so much text on LOC and SOC? In the end it does not add anything to your study, as you even do not use it. You should only describe it in the Discussion.

Reviewer 2 ·

Basic reporting

Introduction is well written. Literature is well referenced and relevant The subject studied is relevant and original.

Experimental design

Research question is well defined. Methods were described with sufficient detail e information to replicate.

Sample and study design

I did not understand the calculation of the minimum sample. The minimum
sample was determined to be 615 children. This value was increased to compensate for possible dropouts estimated at 20% resulting in a sample of 769 (it would not be 738?) five-year-old children. 769 children were the final sample and not minimum sample.

Clinical examination

I suggest clarify the criteria for the diagnosis of tooth wear.

Validity of the findings

Data is robust, statistically sound and controlled.

Results

Table 1:

Some variables don’t add 769 children. I suggest include the missing data in the footnote.

Discussion
I would like to know what the authors think about malocclusion is not associated with the dependent variable (OHRQoL based on the perceptions of children).

Additional comments

I accept the manuscript with minor changes.

---

## Round 0.2 · Minor Revisions

I concluded that you took the comments of the referees very seriously and attempted to include them all in the revised version. The paper improved a lot after revision.

At this stage staff have re-checked the manuscript for plagiarism. As I mentioned earlier there was overlap with a published thesis. This you have addressed in the new version. However, now we found a (new) 9% overlap with a recently published paper in Acta Odont Scand. Parts of the M&M section are the same. I send you a copy of the iThenticate report to indicate where you need to paraphrase or quote and cite sources.

Please change all former changes from red to black and make all new changes in red.

Reviewer 2 ·

Basic reporting

Introduction is well written. Literature is well referenced and relevant The subject studied is relevant and original.

Experimental design

Research question is well defined. Methods were described with sufficient detail e information to replicate.
I understand the calculation of the final sample now and the criteria for the diagnosis of tooth wear is clear too.

Validity of the findings

Data is robust, statistically sound and controlled.
A footnote has been added to Table 1 to clarify the missing answers.
The authors included some information explaining why malocclusion was not associated with the children’s perceptions regarding OHRQoL,

Additional comments

I accept the article without alteration

---

## Round 0.3 · accepted · Accept

The editorial office will further inform you regarding production tasks.